# Genetic Diversity and Association Analysis for Carotenoid Content among Sprouts of Cowpea (*Vigna unguiculata* L. Walp)

**DOI:** 10.3390/ijms23073696

**Published:** 2022-03-28

**Authors:** Frejus Ariel Kpedetin Sodedji, Dahye Ryu, Jaeyoung Choi, Symphorien Agbahoungba, Achille Ephrem Assogbadjo, Simon-Pierre Assanvo N’Guetta, Je Hyeong Jung, Chu Won Nho, Ho-Youn Kim

**Affiliations:** 1Smart Farm Research Center, Korea Institute of Science and Technology (KIST), Gangneung 25451, Korea; frejusariel@gmail.com (F.A.K.S.); dahye0507@kist.re.kr (D.R.); jaeyoung.choi@kist.re.kr (J.C.); jhjung@kist.re.kr (J.H.J.); cwnho@kist.re.kr (C.W.N.); 2Division of Bio-Medical Science and Technology, KIST School, Korea University of Science and Technology (UST), Daejeon 34113, Korea; 3Non-Timber Forest Products and Orphan Crop Species Unit, Laboratory of Applied Ecology (LEA), University of Abomey-Calavi (UAC), Cotonou 05 BP 1752, Benin; agbasympho@gmail.com (S.A.); assogbadjo@gmail.com (A.E.A.); 4West Africa Center of Excellence in Climate Change Biodiversity and Sustainable Agriculture (CEA-CCBAD), Biosciences Research Unit, University Felix Houphouet-Boigny, 22 BP 582 Abidjan 22, Abidjan 582, Côte d’Ivoire; nguettaewatty@yahoo.fr

**Keywords:** biofortification, cowpea, carotenoid, genomics, grain legume, QTL

## Abstract

The development and promotion of biofortified foods plants are a sustainable strategy for supplying essential micronutrients for human health and nutrition. We set out to identify quantitative trait loci (QTL) associated with carotenoid content in cowpea sprouts. The contents of carotenoids, including lutein, zeaxanthin, and β-carotene in sprouts of 125 accessions were quantified via high-performance liquid chromatography. Significant variation existed in the profiles of the different carotenoids. Lutein was the most abundant (58 ± 12.8 mg/100 g), followed by zeaxanthin (14.7 ± 3.1 mg/100 g) and β-carotene (13.2 ± 2.9 mg/100 g). A strong positive correlation was observed among the carotenoid compounds (*r* ≥ 0.87), indicating they can be improved concurrently. The accessions were distributed into three groups, following their carotenoid profiles, with accession C044 having the highest sprout carotenoid content in a single cluster. A total of 3120 genome-wide SNPs were tested for association analysis, which revealed that carotenoid biosynthesis in cowpea sprouts is a polygenic trait controlled by genes with additive and dominance effects. Seven loci were significantly associated with the variation in carotenoid content. The evidence of variation in carotenoid content and genomic regions controlling the trait creates an avenue for breeding cowpea varieties with enhanced sprouts carotenoid content.

## 1. Introduction

A balanced and healthy diet is a global priority, especially in the low-income and developing countries where hunger and malnutrition are more widespread [1,2,3]. Hidden hunger or micronutrient deficiency affects more than two billion people worldwide, causing mental impairment, poor health, low productivity, and death [4,5]. In sub-Saharan Africa, vitamin-A deficiency, for instance, constitutes a serious public health and one of the major causes of blindness and mortality among children under 5 years and pregnant women [6,7,8]. Biofortification as a strategy for increasing nutrients in food plant matrices is advocated to alleviate the high burden of micronutrient deficiency in these regions [9,10].

Gain legumes, including cowpea (*Vigna unguiculata* L. Walp), are important sources of micronutrients and amino acids, exceeding or complementing the profiles of cereals, making them perfect target crops for addressing the global micronutrients deficiency [11,12]. Carotenoids are among the micronutrients found in legumes, and they have a range of benefits for humans [13]. They contribute to the human antioxidant defense system, and their consumption has been proved to improve cellular differentiation and reproduction and to reduce risks of vision impairment, cancer, cardiovascular diseases, and infant mortality [14,15]. The carotenoids found in legume crops include α-carotene and β-carotene, and their hydroxylated forms (lutein, zeaxanthin, violaxanthin, etc.) [16]. Although carotenoids are all composed of isoprene units [17], they differ in their structural and functional properties. Lutein and zeaxanthin possess hydroxyl group at both ends of the molecule, which distinguishes them from β-carotene [18]. β–carotene has β-ionone groups at the two ends, which represents a distinctive feature but also gives the molecule its ability to be converted to retinol, a precursor of vitamin-A, after consumption [19]. Unlike lutein, β-carotene and zeaxanthin belong to the same branch on the carotenoid biosynthesis pathway, with zeaxanthin being the substrate or derivative of β-carotene [20].

Over the past decades, there have been a lot of efforts invested in the biofortification of carotenoids, especially provitamin-A carotenoids, in maize, cassava, and sweet potato, with limited attention paid to grain legume crops [21,22,23]. Cowpea is an important grain legume on which millions of people depend for their daily nutrients needs in the tropics [24,25,26]. It is a model grain legume for genomic studies [27] because of its relatively small (~640.6 Mb) diploid (2*n* = 2*x* = 22 Chromosomes) genome [28], and a large portion of the cowpea germplasm is conserved and easily accessible for research at IITA and USDA GRIN gene banks [27,29,30,31]. Low carotenoid profiles are found in cowpea, dominated (70%) by lutein [32]. However, knowledge about carotenoid biofortification in cowpea is scanty. The level of carotenoid content in cowpea (0–0.1 mg/100 g) [33,34,35,36,37] is lower than the reported values in seeds of other legume grain crops, including chickpeas (0.8–3 mg/100 g) and peas (0.06–2.8 mg/100 g) [38]. Previous research showed the presence of genetic structuration among cowpea germplasm [29,30,39], which constitutes an actionable potential for exploring the natural variation of carotenoid in cowpea. Such information will guide the selection of candidate genotypes and effective methods for increasing carotenoid in cowpea, thereby contributing to the advancement of cowpea biofortification research, which has only focused on zinc and iron [40]. 

Genome-wide association studies (GWAS) has been proven to be a cost-effective, time-saving, and powerful tool for genetic dissection of complex traits [41]. GWAS involves testing for association between each genotypic marker and a phenotype of interest that has been scored across a large number of individuals [42]. It provides a valuable first insight into trait architecture for subsequent validation, which enables detecting rare variants of large effect or common variants of small effect for complex traits [42]. GWAS has been applied to identify genomic regions or quantitative traits loci (QTLs) controlling carotenoid content in some important food crops, including soybean [43], chickpeas [44], maize [45], and cassava [46]. These studies reported that carotenoid content is a complex trait, influenced by various genes with additive and dominance effects [47,48]. To date, there has been no report of quantitative trait loci (QTL) controlling carotenoids biosynthesis in cowpea. The identification of loci associated with carotenoid content in cowpea will help in deciphering the architecture of the trait. 

Sprouts of grain legume or pulses have been shown to increase carotenoid content and overall antioxidant profiles, as well as to minimize the anti-nutritional factors of the dry grains [12,49,50,51]. Recently, we assembled and genotyped a cowpea germplasm collection using the diversity array technology (DArT) sequencing [31]. Here, we assessed the variation of carotenoids, including lutein, zeaxanthin, and β-carotene contents, among sprouts of a set of accessions from the germplasm collection and identified quantitative trait loci (QTLs) associated with carotenoid biosynthesis to boost carotenoid biofortification in cowpea. 

## 2. Results

### 2.1. Variation among Accessions for Sprouts Carotenoid Content

Carotenoid contents were quantified from sprouts of 125 cowpea accessions (Appendix A). The variation in sprouts lutein, zeaxanthin, and β-carotene contents among 125 cowpea accessions is summarized in Table 1. Significant differences (*p* < 0.001) were observed among accessions for the carotenoid compounds. Lutein content varied from 3.7 mg/100 g in sprouts of accession C017 to 182.4 mg/100 g in accession C044, with an average value of 58 mg/100 g (Table 1 and Figure 1). Similarly, sprouts of accessions C017 and C044 recorded, respectively, the lowest (2.2 mg/100 g) and highest (65.2 mg/100 g) values for zeaxanthin content (Table 1 and Appendix A). Sprouts of accessions C113 and C017 had the highest (39.3 mg/100 g) and lowest (2.0 mg/100 g) β-carotene content, respectively, with an average value of 13.2 mg/100 g. Lutein content had the highest average (58 mg/100 g) across accessions, contributing to 67.5% of all carotenoids assessed, which is approximately 4-fold of the values of zeaxanthin and β-carotene (Table 1). 

There were variations in carotenoids content among sprouts following the origins of the accessions (Table 2). The largest variation in lutein content was obtained in sprouts of accessions from north Africa (106.5 ± 50.9 mg/100 g), followed by sprouts of accessions from east Africa (70.4 ± 45.5), Asia (53.8 ± 36.7 mg/100 g), and west Africa (51.0 ± 34.4 mg/100 g). Sprouts of accessions from east Africa recorded the highest zeaxanthin content (17.5 ± 12.1 mg/100 g), followed by sprouts of accessions from Asia and west Africa, while accessions from north Africa showed the lowest content. β-carotene content was the highest in sprouts of accessions from east Africa (14.2 ± 9.1 mg/100 g), followed by sprouts of accessions from Asia (13.4 ± 8.5 mg/100 g) and west Africa (13.0 ± 7.1 mg/100 g). Similar variations were also obtained in β-carotene content among accessions from these three regions. 

### 2.2. Segregation of the Cowpea Accessions into Subgroups Based on the Carotenoid Profiles of the Sprouts

A strong significant positive correlation (*r* ≥ 0.87) was observed among the quantified carotenoid compounds (Appendix A). Hence, we assessed the most appropriate method for grouping the accessions based on the variation observed in the sprout’s carotenoids contents. Among the different combinations of dissimilarities matrices and clustering methods, the combination of Euclidean distance matrix and neighbor-joining algorithm had the highest cophenetic correlation coefficient value (CCC = 0.85) and was used for grouping the accessions (Table 3). The phylogenetic tree built using this combination grouped the accessions into three clusters (Figure 2 and Appendix A).

Figure 2 shows the segregation of the accessions into subgroups based on their sprout carotenoid contents, displayed as color gradient around the phylogenetic tree. Cluster 1 comprising accession C044 and Cluster 2 composed of 17 accessions are characterized by accessions with high carotenoid content (Figure 2). Cluster 3, on the other hand, had the largest size (107 accessions), and it was further divided into two sub-groups: one sub-group made up of accessions with moderate carotenoid content (41 accessions) and the second sub-group characterized by accessions with low carotenoid content accessions (66 accessions).

### 2.3. Genetic Structuration and Linkage Disequilibrium in the Cowpea Germplasm 

A total of 3,120 SNPs (Appendix A) distributed across the 11 chromosomes of cowpea with higher marker densities on chromosomes 3 and 7 (Appendix A) were used to assess the genetic structuration among the cowpea germplasm. Significant genetic differentiation (Fst = 0.25, Table 4) was observed in the germplasm, suggesting the presence of population structure characterized by extensive gene flow (Nm = 5.1) between subgroups. In line with this, the structure analysis identified three subgroups, or clusters (Figure 3), in the germplasm. Cluster 1 had the highest number of accessions (40 accessions), followed by Cluster 2 (37 accessions) and Cluster 3 (32 accessions), with the rest of the accessions (14 accessions) in admixture (Appendix A).

Results of DAPC analysis confirmed the grouping of the 125 accessions into three clusters (Figure 4), as identified by the curve of Bayesian information criterion (BIC) values versus the number of clusters (Appendix A). The biplot based on the two detected linear discriminant axes (DA) (Figure 4), which explained 86.2% and 13.8% of variation in the data, assigned 33 accessions to Cluster 1, 50 accessions to Cluster 2, and 42 accessions to Cluster 3. Cluster 1 had the largest variation for lutein (69.5 ± 41.0 mg/100 g), zeaxanthin (17.1 ± 8.6 mg/100 g), and β-carotene (14.8 ± 8.7 mg/100 g), followed by Cluster 2 and Cluster 3 (Figure 5). 

Furthermore, we examined the pattern of linkage disequilibrium (LD) across the genome (Appendix A). LD was measured as the squared allele frequency correlations (*r*^2^) between pairs of markers. The results showed that 11.4% (11,032) of pairs of comparisons among 1957 markers with minor allele frequency of less than 0.1 were significantly (*p <* 0.01) linked. The average *r*^2^ value across the genome was 0.47. The plot of LD estimates against the physical distance between markers across the genome depicted a persistent LD, which decayed below the critical *r*^2^ = 0.55 at a distance ~1.4 Mbp (Appendix A). 

### 2.4. Analysis of Loci Associated with Carotenoid Biosynthesis 

Out of the 3, 120 loci tested, 7 (Figure 6 and Table 5) showed significant association [3.06 ≤ −log_10_ (*p)* ≤ 4.09] with the variation in carotenoid content among the cowpea sprouts. These loci were distributed on chromosome 6 (*S_Vung_CA1511*, *S_Vung_CA1513*, and *S_Vung_CA1519*), chromosome 7 (*S_Vung_CA1838* and S_Vung_CA1840), chromosome 8 (*S_Vung_CA2146*), and chromosome 11 (*S_Vung_CA3031*). The loci explained 10.10 to 13.51% of the variation of carotenoid content, with locus *S_Vung_CA1840* showing the largest effect (13.51%) for the variation of β-carotene contents among cowpea sprouts (Table 5). 

The identified loci showed pleiotropic effects influencing more than one carotenoid compound, except for *S_Vung_CA1838*, which was solely associated with β-carotene (Table 5 and Figure 7). Loci *S_Vung_CA1511* and *S_Vung_CA1513* were associated with all assessed carotenoid compounds. Loci *S_Vung_CA1519*, *S_Vung_CA2146*, and *S_Vung_CA3031* were associated with the variation in lutein and zeaxanthin contents, whereas locus *S_Vung_CA1840* was associated with both lutein and β-carotene. Both additive and dominance genetic effects were important; however, the dominance effects were higher than additive effects for all significant loci (Table 4). The favorable alleles of the seven loci were the most common alleles contributing to the increasing accumulation of the screened carotenoid compounds in the cowpea sprouts (Appendix A).

## 3. Discussion

### 3.1. Genetic Diversity among the Cowpea for Sprouts Carotenoids Contents 

Genetic diversity in widely consumed food plant species is important to address global food and nutrition security challenges [52]. In this study, we assessed the profiles of carotenoids, including lutein, β-carotene, and zeaxanthin, in sprouts of 125 cowpea accessions. Significant variations were observed among sprouts of the cowpea accessions for the contents of lutein (3.7 to 182.4 mg/100 g), zeaxanthin (2.2 to 65.2 mg/100 g), and β-carotene (2.0 to 39.3 mg/100 g), suggesting that there is genetic variability in the germplasm for enhancing carotenoids biosynthesis. The results, in line with previous findings [32], confirmed that lutein is the most abundant (~70%) carotenoid compound in cowpea. Hence, the biofortification of carotenoids in cowpea can be very beneficial due to the critical role of lutein in human vision, immunity, and anti-inflammatory system [18]. In the present study, we found evidence of variability in carotenoid content following the origins of the cowpea accessions. Sprouts of accessions from major cowpea growing regions, including Africa and Asia, exhibited significant variations in the screened carotenoid compounds, especially in lutein content. This also showed that there is a great potential at the regional level that can be harnessed for carotenoid biofortification, which could be of great interest considering the high consumption of cowpea and the need for food fortification in these regions [10,26]. The average contents of lutein (58.0 mg/100 g), β-carotene (13.2 mg/100 g), and zeaxanthin (14.7 mg/100 g) in the 5-day-old cowpea sprouts were higher than the values reported in cowpea seeds [35], as well as in the 2-day-old average contents of cowpea sprouts [37]. Such changes are expected, since carotenoids, as photosynthetic plants’ pigments, can increase in content upon germination and plant growth as a result of expansion of chloroplast tissues, the main source of carotenoids [53]. These findings strongly suggest that producing and promoting cowpea sprouts, especially five-day-old sprouts-based diets, can help prevent carotenoid deficiency, which is more prevalent in areas where cowpea is a staple food crop [32,54]. Some of the accessions used in this study, including C044, C113, C115, C097, and C095, showed high profiles of carotenoid content and could be used for the purpose. 

There has been limited research on carotenoid biofortification in cowpea. The current study is very promising because it established carotenoid profiles of sprouts in a wide range of cowpea germplasm. The analysis of the relationship among accessions based on the carotenoid content in their sprouts showed the presence of subgroups, indicating there are possibilities of hybridization between accessions for increased carotenoid biosynthesis. Furthermore, the high positive correlation among the profiles of the screened carotenoid compounds suggests that they can be improved concurrently. The level of zeaxanthin was, in general, higher than β-carotene content. This can be attributed to the conversion of β-carotene into zeaxanthin. The correlation between these compounds (β-carotene and zeaxanthin), although positive, was the lowest among the three pairs of comparisons among compounds (Appendix A), confirming the possible oxygenation of β-carotene to form zeaxanthin.

### 3.2. Prospects of Marker-Assisted Selection for Nutrient Enhanced Cowpea Sprouts 

Carotenoid content in plants is an important trait for human health and nutrition [47,55]. The discovery of markers associated with this trait can help in fast tracking, selection, and breeding of cowpea varieties with high carotenoid content in the sprouts. The results showed that the 125 accessions can be distributed into three groups, which corroborates our previous findings in the population of origin of the accessions [31]. Although the presence of population structure within the germplasm is a positive indication of genetic potential for improvement, it can be a confounding factor in testing the associations between markers and phenotypic variation [56,57]. This was considered in the GWAS model used. 

The analysis of association between loci revealed a strong LD pattern in the germplasm, as reflected by the high mean of correlations between (*r*^2^ = 0.47) pairs of markers across the genome. Notably, high LD decay distance (1.4 Mbp) was obtained in the germplasm, which is within the range of ~500 kb to 1.88 Mb, previously reported in *V. unguiculata* subspecies [58,59]. This further suggests that there is a possibility of detecting genetic markers associated with carotenoid biosynthesis in the germplasm [60]. In line with that, the genome-wide association analysis identified seven major loci (R^2^ > 10%) [61] that significantly explained the variation in carotenoid content among the cowpea sprouts. These results suggest that carotenoid content in cowpea is a polygenic trait [62], and the identified markers can support marker-assisted selection. These markers showed pleiotropic effects, suggesting their usefulness in selection of parental lines with high sprouts carotenoids content. The pleiotropic effect also substantiated that screened carotenoid compounds belong to the same pathway and may be under the influence of a similar genes network. Furthermore, both dominance and additive gene effects were important in explaining the observed variation in the content of the specific carotenoid compounds among sprouts of the cowpea germplasm, with a high influence of the dominance effects over the additive effects. High influence of dominance gene effects in carotenoid biosynthesis was also reported in the African marigold [63]. Similarly, Kandianis, et al. [64] showed that the variation of β-carotene content in maize was fully explained by additive and dominance gene effects, rather than additive genes effects only. These results demonstrate that carotenoid content is a quantitative trait, and there are chances of heterosis and/or transgressive segregation, resulting in increased content in sprouts of progenies of crosses among superior individuals [63,65]. 

Sprouts of grain legume crops can be improved through varietal selection [66]. There has been extensive research conducted on soybean sprouts improvement, including breeding and genetic improvement [66,67,68]. To our knowledge, this is the first study reporting on genomic regions associated with carotenoid content in cowpea sprouts. Hence, the identified markers will be very useful and can be validated in a biparental population for effective use in a breeding program. Since low genetic diversity was observed in the germplasm, the use of recombinant inbred lines (RILs) population may be more appropriate for validation [69]. The use of RILs population can also help to broaden the genetic variation of the trait amenable for the discovery of more quantitative trait loci [70]. Additionally, testing the effectiveness of these loci in sprouts of cowpea seeds from replicated trials in time and space can be more informative to account for the interaction effects of quantitative trait loci and environments on carotenoid content, as well as other characteristics, such as seed size, seed coat, and color, that can influence sprouts yield and quality [66,71].

## 4. Materials and Methods

### 4.1. Plant Materials

The cowpea diversity panel comprises 125 accessions from Africa (93 accessions), Asia (29 accessions), America (2 accessions), and 1 Oceania (1 accession) (Appendix A). Seeds of the accessions were obtained from a cowpea collection of the Laboratory of Applied Ecology (LEA) of the University of Abomey-Calavi (UAC; Abomey-Calavi, Benin). Ethanol analytical grade and sodium hypochlorite were purchased from Sigma Aldrich, Seoul, Korea. The seeds were surface sterilized with 70% Ethanol (*v*/*v*) for 30s and 0.5% sodium hypochlorite (NaOCl) solution for 2 min and washed twice with distilled water. Seeds were soaked in distilled water for 12 h and germinated in sprouts plastic trays (Appendix A) in a growth chamber, 16h of light and 8 h of dark cycle. The temperature and relative humidity (RH) in the growth chamber was kept at 24 ± 1 °C and 80%, respectively. The seeds were moistened with distilled water every 8 h for 5 days. Sprouts were harvested 5 days after germination. Sprouts samples were frozen in liquid nitrogen and freeze dried overnight for carotenoid analysis. 

### 4.2. Carotenoids Profiling 

#### 4.2.1. Sample Extraction 

Profiles of carotenoids, including lutein, zeaxanthin, and β-carotene in the sprouts of each cowpea accession, were assessed. Standards of lutein, zeaxanthin, and β-carotene and reagents/solvents, including acetone, hexane, ethanol (ETOH), methanol (MeOH), potassium hydroxide (KOH), and Methyl tert-butyl Ether (MTBE), were purchased from Sigma Aldrich, Seoul, Korea. Fifty milligrams (50 mg) of finely ground freeze-dried sprouts samples were extracted with 50 μL of 1 N potassium hydroxide in 1 mL mixture solution of Acetone/Ethanol/Hexane (1:1:2, *v*/*v*/*v*). The mixture was vortexed for 20 s, sonicated (Vibra-cell™, Sonics, Newtown, CT, USA) at 40 °C for 30 min, and centrifuged (Labogene 1248 R, Seoul, Korea) for 5 min, 1200 rpm at 4 °C. The upper hexane layer of each sample was collected and passed through a membrane filter (PVDF syringe filter, hydrophobic, 13 mm diameter, 0.22 μm pore size, Whatman International, Maidstone, UK). The extraction process was repeated twice, and the resulting extracts were mixed and passed through a stream of nitrogen gas for removal of the hexane using nitrogen evaporator (Allsheng MD 200, Hangzhou Allsheng Instrument Co., LTD, Hangzhou, China). The extracts were dissolved in acetone for high-performance liquid chromatography (HPLC) analysis.

#### 4.2.2. HPLC Analysis of Carotenoids 

Carotenoids were analyzed in a Dionex ultimate 3000 LC machine equipped with a standard auto-sampler, a binary gradient pump, and a variable wavelength detector (VAD). Specific carotenoid compounds were separated on a reverse phase C30 YMC carotenoid column, (5 µm, 250 × 4.6 mmL.D.mm) using mobile phases consisting of MTBE: MeOH (90:10 *v*/*v*, solvent A) and MeOH: H_2_O (95:5 *v*/*v*, solvent B) in a linear gradient. The gradient elution was 20% A and 80% B from 0 to 7 min, followed by 25% A and 75% B to 15–25 min, 100% A to 40 min, and 20% A and 80% B to 45–50 min. The flow rate was 0.7 mL/min, and the column temperature was maintained at 35 °C. The eluting peaks were monitored at 450 nm wavelength (Appendix A). Lutein, β-carotene, and zeaxanthin contents were estimated based on the calibration curve of lutein standard (Y = 48.9X + 2.2, R^2^ = 0.997), β-carotene standard (Y = 30.3X − 0.7, R^2^ = 0.999), and zeaxanthin (Y = 12.2X − 0.4; R^2^ = 0.999), respectively.

#### 4.2.3. Data Analysis 

All measurements were performed in triplicates. Analysis of variance of the performance of the cowpea accessions for sprouts carotenoids contents was computed in the agricolae R package [72]. Means were separated using the Tukey’s honestly significant difference (*α* = 0.05) in the car R package [73]. Correlation analysis among the parameters was performed in factoextra R package [74]. 

Clustering analysis was performed to assess the relationship among accessions based on the carotenoid content in their sprouts. For this purpose, we computed the cophenetic correlation coefficient (CCC) of different combinations of dissimilarities matrices (Euclidean and Manhattan) and clustering methods (Ward. D, neighbor-joining, and the unweighted pair group method with arithmetic mean method). The combination with the highest CCC value was used to perform the hierarchical clustering analysis, and the optimal number of clusters was inferred in the NbClust R package [75]. The resulting phylogenetic tree was exported using the ape R package [76] for graphical annotation and annotation, with the contents of lutein, zeaxanthin, and β-carotene in the sprouts of each accession depicted around the phylogenetic tree using the graphical phylogenetic analysis (GraPhlAn v1.1.4) [77].

### 4.3. Genome-Wide Association Studies (GWAS) 

The genomic data used in this study consisted of 3,120 SNPs markers (Appendix A), previously reported among a diversity panel of cowpea [31], which includes the 125 accessions used in the present study. The genetic diversity parameters of the germplasm were estimated in GenAlex [78]. To control false positives association, the population structure of the germplasm was assessed. 

#### 4.3.1. Population Structure

A genetic population structure analysis was performed in LEA R package [79]. In this method, the optimal number of clusters (K) is determined using the cross-entropy criterion, which is based on the prediction of a fraction of masked genotypes and cross-validation approach [79]. Ten repetitions were performed for each value of K (K = 1:5) and the optimal K value selected. The membership of the accessions in a specific cluster was depicted using the barplot function, with the critical coancestry coefficient set at 0.55. To confirm the optimal number of clusters in the germplasm, we performed a discriminant analysis of principal components (DAPC) in the adgenet R package [80]. 

#### 4.3.2. Linkage Disequilibrium (LD) Analysis

LD within the cowpea germplasm was estimated in Tassel v5.2.60 [81]. SNPs markers with minor allele frequency above 0.1 were included in the LD analysis. LD was measured as the squared allele frequency correlations (*r*^2^) between pairs of markers across the genome [58]. LD decay pattern was depicted as a function of *r*^2^ along physical distance (kb) in ggplot2 R package [82], where only *r*^2^ with *p <* 0.01 were included. The critical *r*^2^ for LD decay was estimated as the 95^th^ percentile of distribution of the square root transformed of the correlations values between unlinked markers [83].

#### 4.3.3. GWAS Analysis

GWAS was performed using Tassel v5.2.60 [81] and rMPV R package [84]. The phenotypic variation represented herein by the carotenoid content of the accessions was subjected to a rank-based transformation, a method reported to give the best and consistent performance in identifying the causal polymorphism among other transformation approaches [85] in the bestNormalize R package [86]. A general linear model (GLM) approach [57] was used for the association analysis, and the coefficients of coancestry of the accessions were incorporated in the model as covariates (Q matrix) to correct for false positives. Manhattan plots were used for the visualization of the GWAS results. Markers that passed the significance threshold *p <* 0.001 [i.e., −log_10_ (*p*) > 3] were defined as genomic regions or loci associated with carotenoid biosynthesis in cowpea sprouts [87,88]. 

## 5. Conclusions

This study revealed that the level of carotenoid content varied among sprouts of cowpea accessions. The accessions were grouped into three clusters based on their carotenoid contents, with some of them exhibiting high profiles of carotenoids, and they can be recommended for production and promotion of high integration of cowpea sprouts in the daily diet consumption in food-insecure regions. The presence of subgroups in the population was also confirmed by analysis of the genetic structure. However, the germplasm had low genetic diversity, which calls for more research efforts to broaden the genetic basis of cowpea for high carotenoids content, as well as other important characteristics of cowpea sprouting varieties. Seven candidate loci, *S_Vung_CA1511*, *S_Vung_CA1513*, *S_Vung_CA1519*, *S_Vung_CA1838*, S_Vung_CA1840, S_Vung_CA2146, and *S_Vung_CA3031*, were identified to support molecular breeding for sprouting cowpea varieties with enhanced carotenoids contents.

## Figures and Tables

**Figure 1 ijms-23-03696-f001:**
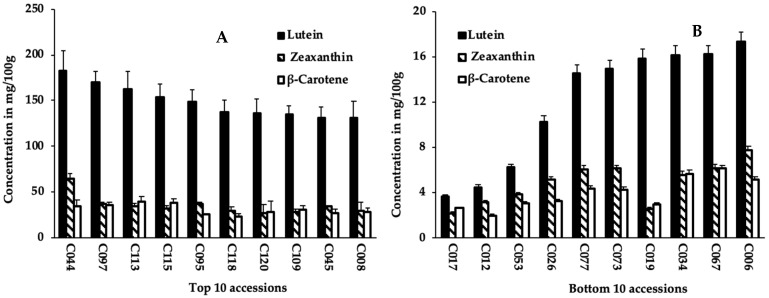
Lutein, zeaxanthin, and β-carotene content of sprouts of the top 10 (**A**); and bottom 10 (**B**) cowpea accessions.

**Figure 2 ijms-23-03696-f002:**
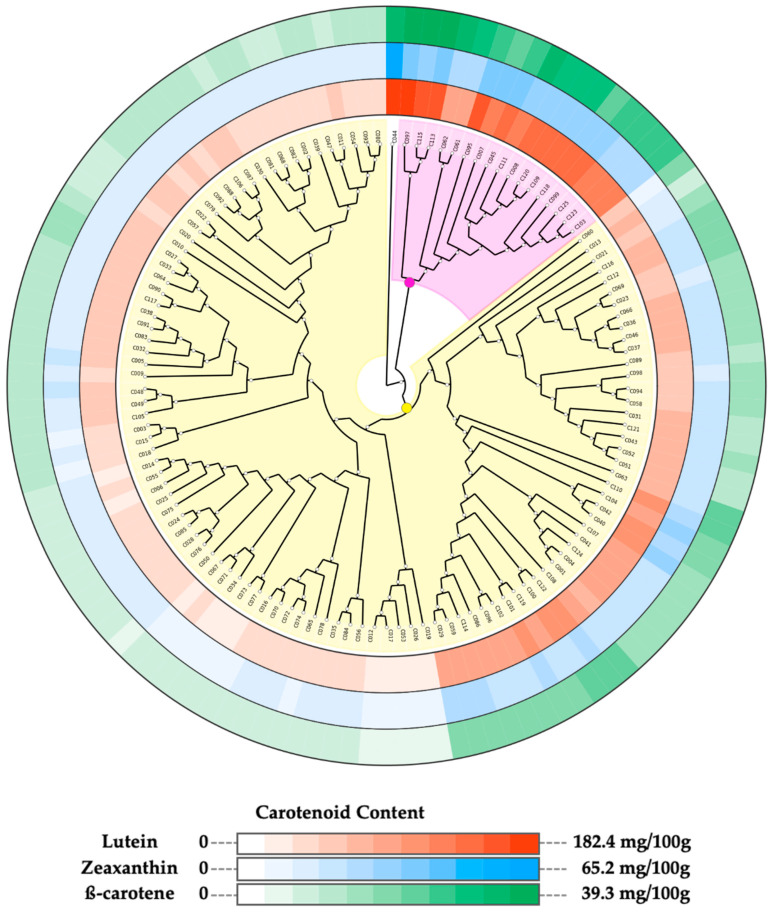
Phylogenetic three of 125 cowpea accessions constructed based on their carotenoid contents. Neighbor-joining clustering divided the accessions into three clusters. Cluster 1 (C044): no background, Cluster 2: light purple background, Cluster 3: yellow background. Concentrations of carotenoids in sprouts of each accession are displayed in color gradient around the tree. From the outermost track: β-carotene, zeaxanthin, and lutein.

**Figure 3 ijms-23-03696-f003:**
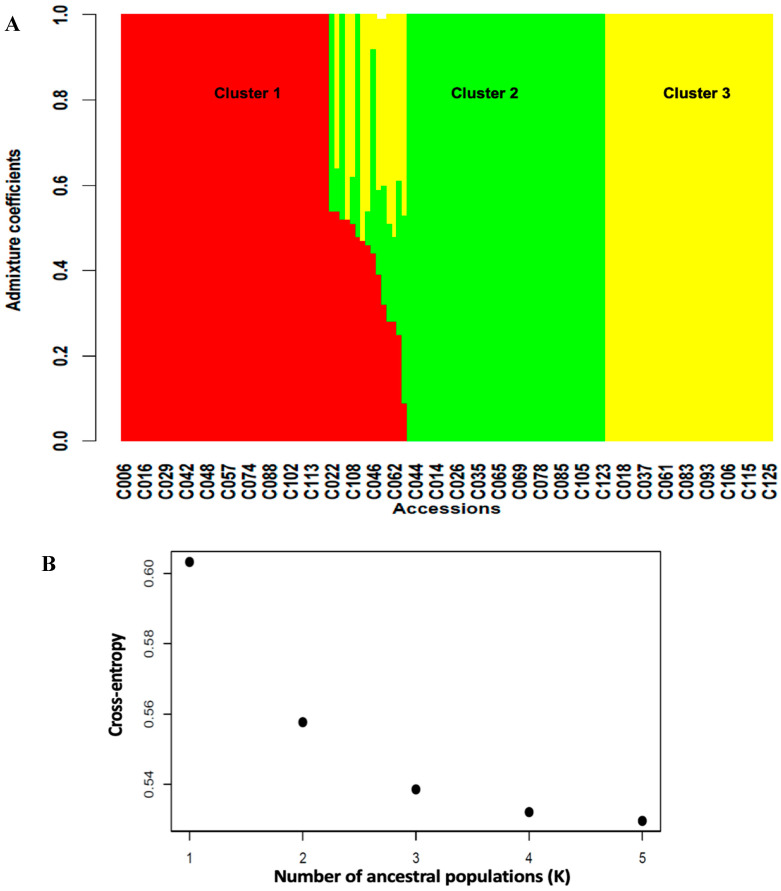
(**A**) = barplot of admixture coefficients of the 125 accessions depicting the population structure in the cowpea-mapping panel. (**B**) = Three clusters were identified based on the curve of cross-entropy versus the number of ancestral populations.

**Figure 4 ijms-23-03696-f004:**
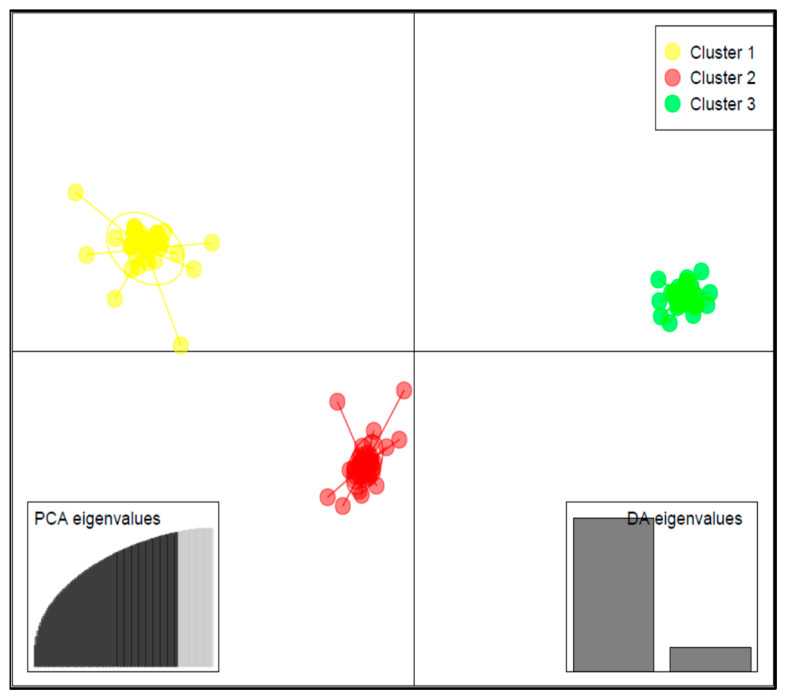
Discriminant analysis of principal component (DAPC) biplot showing the structuration of the 125 cowpea accessions into three subgroups based on the SNP makers.

**Figure 5 ijms-23-03696-f005:**
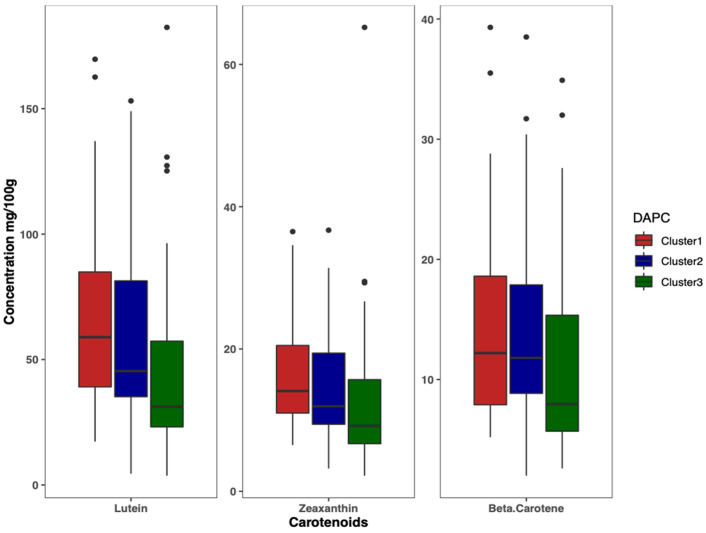
Boxplot showing the variation of carotenoids content among clusters.

**Figure 6 ijms-23-03696-f006:**
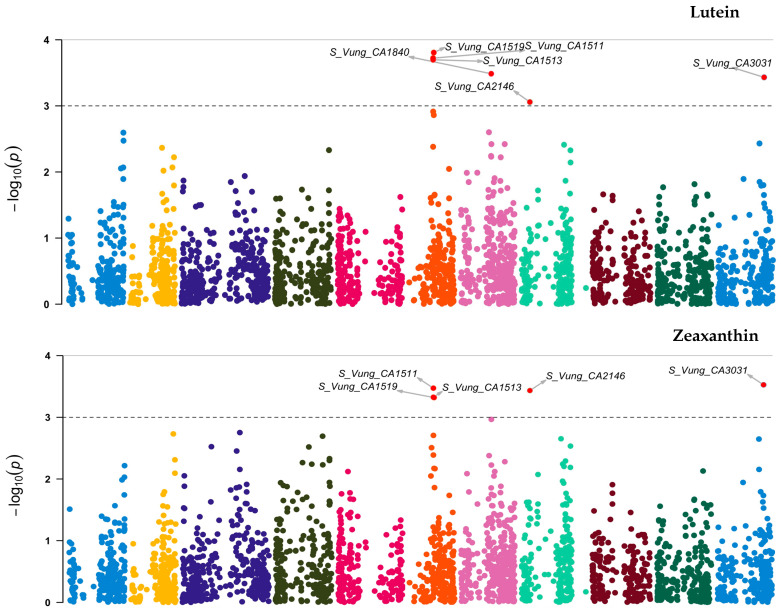
Manhattan plots of loci associated with lutein, zeaxanthin, and β-carotene. Loci are represented by small dot colors according to their localization on 11 chromosomes (Chr 1:11) of cowpea. The red dots above the cut-off value [–log _10_(*p*) *= 3*)] indicate the significant loci.

**Figure 7 ijms-23-03696-f007:**
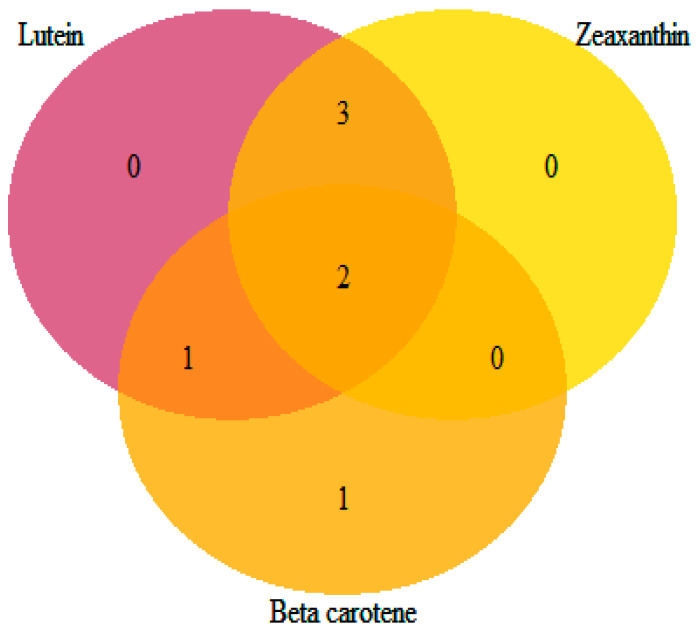
Venn diagram showing the number of loci shared among the targeted carotenoids.

**Table 1 ijms-23-03696-t001:** Descriptive statistics of the variation of lutein, zeaxanthin, and β-carotene among sprouts of 125 cowpea accessions.

Parameters	Min	Max	Mean ± SD	*p*-Value	Tukey’s HSD
Lutein	3.7	182.4	58.0 ± 12.8	<0.001	60.0
Zeaxanthin	2.2	65.2	14.7 ± 03.1	<0.001	18.5
β-carotene	2.0	39.3	13.2 ± 02.9	<0.001	13.2

S = Sprouts, c = mg/100 g; Tukey’s HSD = Tukey’s highly significant difference at α = 0.05.

**Table 2 ijms-23-03696-t002:** Variation of lutein, zeaxanthin, and β-carotene contents following the regions of origin of the cowpea accessions.

Regions	Carotenoids	Means ± SD	Coefficient of Variation	Number of Accessions
Asia	Lutein	53.8 ± 36.7	1.5	29
	Zeaxanthin	13.7 ± 7.9	1.7	
	ß-Carotene	13.4 ± 8.5	1.6	
East Africa	Lutein	70.4 ± 45.5	1.5	33
	Zeaxanthin	17.5 ± 12.1	1.4	
	ß-Carotene	14.2 ± 9.1	1.6	
North Africa	Lutein	106.5 ± 50.9	2.1	3
	Zeaxanthin	11.5 ± 9.9	1.2	
	ß-Carotene	10.7 ± 13.1	0.8	
West Africa	Lutein	51.0 ± 34.4	1.5	57
	Zeaxanthin	13.0 ± 7.1	1.8	
	ß-Carotene	11.6 ± 6.9	1.7	
US–Oceania	Lutein	47.5 ± 35.4	1.3	3
	Zeaxanthin	12.3 ± 6.2	2	
	ß-Carotene	10.6 ± 6.5	1.6	

**Table 3 ijms-23-03696-t003:** Cophenetic correlation coefficient (CCC) between distance matrices and clustering. Algorithms for inferring clusters among cowpea accessions.

CCC	Carotenoid-Content-Based Clustering	
	Manhattan	Euclidean
ward.D	0.71	0.59
UPMGA	0.83	0.84
NJ	0.84	0.85

**Table 4 ijms-23-03696-t004:** Genetic diversity indices among the cowpea accessions using Nei method.

Clusters	Size	Ho	Hs	Fis	Fst	Gst	Nm
Total	125	0.04	0.23	0.84	0.25	0.23	5.71

Ho = Mean Observed Heterozygosity over k clusters, Hs = Mean Expected Heterozygosity He over k clusters, Hs = pop allele frequency, Fis = Inbreeding coefficient within individuals, Fst = genetic differentiation among clusters, Gst = Analog of Fst, adjusted for bias, Nm = gene flow between populations.

**Table 5 ijms-23-03696-t005:** Significant loci associated with carotenoids biosynthesis in cowpea sprouts.

Loci Names	Allele	Chr	Position (kb)	Compounds	R^2^ (%)	−log _10_(*p*)	A_Effect	D_Effect
*S_Vung_CA1511*	G/A	6	18444146	Lutein	12.14	3.72	0.42	1.15
				Zeaxanthin	11.60	3.47	0.51	1.10
				β-Carotene	11.00	3.40	0.47	1.09
*S_Vung_CA1513*	G/A	6	18455640	Lutein	12.06	3.70	0.42	1.12
				Zeaxanthin	11.14	3.33	0.51	1.05
				β-Carotene	11.12	3.31	0.47	1.06
*S_Vung_CA1519*	A/T	6	18955912	Lutein	12.39	3.81	0.58	1.69
				Zeaxanthin	11.12	3.32	0.48	1.56
*S_Vung_CA1838*	C/T	7	22819466	β-Carotene	10.30	3.06	0.55	0.97
*S_Vung_CA1840*	G/T	7	22946212	Lutein	11.42	3.49	0.49	0.95
				β-Carotene	13.51	4.09	0.55	1.02
*S_Vung_CA2146*	T/C	8	6425230	Lutein	10.10	3.06	0.43	0.74
				Zeaxanthin	11.48	3.43	0.43	0.81
*S_Vung_CA3031*	C/T	11	34652559	Lutein	11.25	3.43	0.05	1.11
				Zeaxanthin	11.77	3.53	0.06	1.11

Chr = Chromosome; A_Effect = additive effect; D_Effect = Dominance effect, R^2^ = R-squared for the marker.

## Data Availability

The data presented in this study are available in the Appendix A.

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
