# Peer review of "Genetic Diversity and Association Analysis for Carotenoid Content among Sprouts of Cowpea (Vigna unguiculata L. Walp)"

_ijms, 2022, doi:10.3390/ijms23073696_

Round 1

Reviewer 1 Report

This article should be accepted after minor revision as following;

Introduction
-Paragraph 2: Authors should give more explanation about 3 types of carotenoids.

Results
-Authors should provide more sharpen figure in Figs. 3-6.

Discussion
- "In fact, unlike lutein, β-carotene ... of β-carotene [49]" in lines 251-253 should move to introduction.

Author Response

We thank the reviewer for the comments which helped to refine the manuscript.

Introduction
-Paragraph 2: Authors should give more explanation about 3 types of carotenoids.

We added more explanation about the 3types of carotenoids (see lines 62-69)

Results
-Authors should provide more sharpen figure in Figs. 3-6.

We increase the resolution of figures and also changed the circular Manhattan plot to a rectangular plot, which we believe would improve the quality of the figures.

Discussion
- "In fact, unlike lutein, β-carotene ... of β-carotene [49]" in lines 251-253 should move to introduction.
This was moved to the introduction (lines 67-69)

Reviewer 2 Report

In this manuscript, the authors analyzed the carotenoid profiles of 125 cowpea sprouts. In particular, the authors assessed the potential association between cowpea carotenoid profiles and SNPs in the cowpea genome.

The manuscript constitutes a continuation of a recently published article by the same team, entitled “ Diversity, population structure, and linkage disequilibrium among cowpea accessions”.

The global idea and the aim of the study are of great interest. Also, the approach used is well-conceived. However, how the results are reported in the main text should be improved. The abstract needs rewriting to improve its information content and provide more context and rationale for the study. Indeed, it is important to highlight (and discuss) the results concerning the large variation in carotenoid content among the accessions studied as well as the potential association of this variation with SNPs in the cowpea genome. Some adaptation of writing style is advised, omitting some details, and emphasizing more the major trends and most salient results. English needs a little fine-tuning. Finally, the quality of the figures needs to be improved (not readable). As stated, this is an interesting study, but the text and figures quality need to be improved before it can be accepted for publication in a high-impact journal.

Specific Comments:

I suggest adding figure S1 as an abstract graph following some adjustments.

Abstract:

L18: Reformulate or remove the station “for increased carotenoid among sprouts of cowpea”. The expression reflects a “process” rather than the aim of the study. The aim is to identify potential QTL associated with the carotenoid content, which can be used in future breeding or selection programs of cowpea.

L23_24: Switch the order of the two sentences. Here, I suggest adding information about the C044 accession (have the highest carotenoid content and show up in a unique cluster).

Keywords:

I suggest replacing “food security” with QTL

Introduction:

The introduction section is well presented.

L88: The station “to boost carotenoid biofortification in cowpea” Right meaning than stated in the abstract (L18).

Result

L92: Table S2 not S1.

L93: Remove Figure S1; Remove “performance

L99: Add “,respectively,” after recorded.

L143: Replace " point " with " comma ".

L135: Figure 2: Please provide the decomposable figure of Figures, whose parts are movable and editable.

What about the link between the accession’s carotenoid profiles and cowpea accessions origin (Table S1)? Add the information in the result text and in the first part of the discussion.

L171: Figure 4 B, remove grid and background from plot B (ggplot2) or simply remove them with a graph edit program. Also, Figure quality (pixel) must be improved (not readable). For all figures.

Discussion

L220_238: I recommend reporting here the variation of lutein, β-carotene, and zeaxanthin according to the cowpea accessions (from … to …). Because of this great variation, the biofortification of carotenoids in cowpea is of great potential and interest.

L241_245: Remove “The levels of ………… Hence, there is a need to increase carotenoid content in cowpea.” Not informative here, report in the introduction.

Author Response

We appreciate the comments and suggestions made to improve the manuscript.

I suggest adding figure S1 as an abstract graph following some adjustments. We improved the figure S1 which will be used as graphical as suggested.

Abstract

L18: Reformulate or remove the station “for increased carotenoid among sprouts of cowpea”. The expression reflects a “process” rather than the aim of the study. The aim is to identify potential QTL associated with the carotenoid content, which can be used in future breeding or selection programs of cowpea.

Line 18 was reformulated as follows: ‘’We set out to identify quantitative trait loci (QTL) associated with carotenoid content in cowpea sprouts’’

L23_24: Switch the order of the two sentences. Here, I suggest adding information about the C044 accession (have the highest carotenoid content and show up in a unique cluster).

We switched the order of the sentences and add the information about accession C044 as suggested. (Lines 23-24)

I suggest replacing “food security” with QTL.

This was revised.

Introduction:

The introduction section is well presented.

L88: The station “to boost carotenoid biofortification in cowpea” Right meaning than stated in the abstract (L18).

Result

L92: Table S2 not S1.

We considered both table S1 et table S2, as together they helped to link the origins of the accessions and their carotenoid contents

L93: Remove Figure S1; Remove “performance” This was removed

Figure S1 was removed, however as the Figure S1in the previous version of the manuscript is now proposed as graphical abstract, the in-text citation of the text has also changed. Consequently, Figure S2 becomes Figure S1.

L99: Add “, respectively,” after recorded. This was added, lines 124

L143: Replace " point " with " comma ".  The revision was done.

L135: Figure 2: Please provide the decomposable figure of Figures, whose parts are movable and editable. We provided decomposable figure for all figures.

What about the link between the accession’s carotenoid profiles and cowpea accessions origin (Table S1)? Add the information in the result text and in the first part of the discussion.

This was addressed (see lines 130-140 results section, lines 413-414 and lines 418-424 of the discussion). A table summarizing the variations following the regions of origin of the accessions was added in the result section.

L171: Figure 4 B, remove grid and background from plot B (ggplot2) or simply remove them with a graph edit program. Also, Figure quality (pixel) must be improved (not readable). For all figures.

All figures have been reviewed and added in detachable forms. Figure 4B was edited.

Discussion

L220_238: I recommend reporting here the variation of lutein, β- carotene, and zeaxanthin according to the cowpea accessions (from ... to ...). Because of this great variation, the biofortification of carotenoids in cowpea is of great potential and interest.

This was addressed in lines 413-414 and lines 418-424 of the discussion).

L241_245: Remove “The levels of ............ Hence, there is a carotenoid content in cowpea.” Not informative here, report in the introduction. This was reported in the introduction.

Round 2

Reviewer 2 Report

I appreciate the authors' work to improve the quality/readability of the manuscript. 

Overall, corrections/suggestions are well done. Only for supplementary material, please check the version - it contains the old version of Figure S1, which should be included as a graphical abstract.

Finally, I congratulate the authors for this work.
I recommend that this article be accepted. 

Author Response

Dear Reviewer, 

Overall, corrections/suggestions are well done. Only for supplementary material, please check the version - it contains the old version of Figure S1, which should be included as a graphical abstract.

Figure S1 was removed from the Supplementary material, and the numbering revised in both the supplementary material and the main manuscript.

Thank you 

Regards,